# The Relationship between Family Functioning, Emotional Intelligence, Loneliness, Social Support, and Depressive Symptoms among Undergraduate Students

**DOI:** 10.3390/bs14090819

**Published:** 2024-09-14

**Authors:** Mimi Zhao, Nor Ba’yah Abdul Kadir, Muhammad Ajib Abd Razak

**Affiliations:** Faculty of Social Sciences and Humanities, Universiti Kebangsaan Malaysia, Bangi 43600, Malaysia; p114390@siswa.ukm.edu.my (M.Z.); muhdajib@ukm.edu.my (M.A.A.R.)

**Keywords:** family functioning, emotional intelligence, loneliness, social support, depressive symptoms among college

## Abstract

The transition from adolescence to college is a significant developmental stage marked by challenges such as high academic pressure, changes in living environments, and social support systems. These challenges can lead to increased rates of depression among college students. This study investigates the relationships between family functioning, emotional intelligence, loneliness, social support, and depressive symptoms in Chinese college students. A cross-sectional design was employed, with data collected via questionnaires from undergraduate students at Xi’an Jiaotong University. Variables such as family functioning, emotional intelligence, loneliness, social support, and depressive symptoms were assessed. Statistical analyses, including correlation and hierarchical regression, were conducted to explore these relationships. The study found a diverse distribution of depressive symptoms among students, with significant gender differences indicating higher depression rates in females. No significant differences were observed across academic disciplines, grades, or between only children and non-only children. Family functioning, emotional intelligence, and social support were negatively correlated with depressive symptoms, while loneliness was positively correlated. Hierarchical regression analysis confirmed that social support significantly moderated the relationship between family functioning and depressive symptoms. Mediation analysis showed that emotional intelligence and loneliness mediated this relationship. The findings highlight the complex interplay between family dynamics, emotional capabilities, social connectedness, and mental health. Enhancing family functioning, emotional intelligence, and social support can effectively reduce depressive symptoms among college students. These results underscore the need for holistic interventions that address multiple aspects of students’ social and emotional lives.

## 1. Introducing

The transition from adolescence to college is a critical developmental stage towards adulthood, marked by diversity, mobility, and complexity in an individual’s life experiences [1]. College students are more prone to depression compared to the general population or students who do not attend college [2]. This is due to their physical and psychological transition from late adolescence to early adulthood, significant changes in their living environment and interpersonal relationships, high academic pressure, and grim employment prospects [3]. Various habitual factors associated with a new environment can also lead to feelings of distress and irritability [4], presenting challenges such as living independently, learning independently, and accessing social support [5,6].

Students often describe the transition to university as a period when they are nostalgic for home [7]. In China, the prevalence of one-child families from 1982 to 2015 [8]. exacerbates the impact of living away from their parents. Universities require students to live on campus with roommates, which negatively affects students’ mental health due to the loss of social support [9]. A survey showed a 27.13 percent detection rate of maladjustment among 5818 college freshmen in China, particularly among women, only children, and students from low socioeconomic status families [10]. Semi-structured interviews with emerging adults highlighted that solitude can lead to both positive experiences, like concentration and freedom, and negative experiences such as loneliness and rumination [11].

Leaving home for college often increases the risk of social isolation [12,13,14], a trait predicting poor mental health outcomes among students. The COVID-19 pandemic and the shift to online learning have intensified feelings of loneliness and isolation [15]. Students reported a lack of connection and increased mental health challenges due to the virtual learning environment [13,16]. Continuous isolation and lack of interaction with peers and teachers during abrupt online class transitions have psychological effects on students [15].

Smartphone and social media use is prevalent among Chinese college students [17], and extensive use is associated with poorer adjustment outcomes for freshmen [18]. Internet and smartphone addiction can exacerbate loneliness [19], which in turn can lead to depressive symptoms and further addiction [20]. High levels of mobile phone dependence negatively affect mental health, leading to problems like depression and anxiety [21]. Depressed students, especially those in humanities and arts, are more likely to become addicted to the Internet [22]. Depressive symptoms also increase the risk of suspension or withdrawal from school [23] and can lead to severe consequences like suicide [24].

Emotional intelligence significantly impacts life satisfaction in emerging adults, which is a crucial issue in their identity development [25]. Due to China’s exam-oriented education system, many college students receive little training in emotional intelligence, which is positively correlated with subjective well-being and negatively with stress and social anxiety [26,27]. An inability to cope with this transition can lead to maladjustment, affecting student well-being and mental health [28,29,30], leading to issues like sleep disorders [31], academic problems [32,33,34], and decreased social support [29]. A longitudinal study of 843 Chinese freshmen showed significant increases in emotional adjustment, a stable academic adjustment, and a significant decline in social adjustment during the first year [35]. Therefore, students need to make significant social and emotional adjustments [34].

Since the COVID-19 outbreak, severe psychopathology has been reported [36,37]. Psychoanalysts believe pandemics activate primitive human defenses [38]. The epidemic’s impact on mental health problems like depression, anxiety, and insomnia will be long-lasting [39]. Various countries implemented measures like lockdowns, school closures, and social distancing to curb the virus spread [40,41]. Studies showed decreased COVID-19 transmission rates due to reduced mobility [42], and school closures reduced transmission and mortality in the student population [43]. However, prolonged embargoes can cause serious damage to physical and mental health and have adverse social and economic effects [44,45,46]. As a result, many countries eased restrictions gradually, while China maintained strict measures [40]. Chinese students experienced dramatic changes in their education, social life, and mental health, and especially increased isolation [15]. Longer isolation periods exacerbate pre-existing psychological and psychiatric problems [47,48,49,50].

Loneliness is more intense for people living alone and young people [51]. The shift to online learning during the pandemic increased students’ distress [52]. Online socialization also led to greater loneliness [53], with concerns about maintaining relationships through video chat increasing fear and loneliness [54]. A study in the Philippines showed widespread loneliness among college students during the pandemic [15]. Higher levels of loneliness affect students’ health and academic performance [55]. Lack of social contact and feeling lonely predict depressive symptoms [56].

The closure of education centers and home confinement increased violence, poverty, overcrowding, and technology misuse within families [57]. Students were concerned about their health and that of loved ones during COVID-19 [58]. Health professionals are at high risk for mental health problems due to fear of spreading the virus [59]. Women tend to be more aware of the pandemic and suffer more psychological distress due to information overload [51]. Family function is a protective factor for mental health [60].

Social support is crucial during the initial transition from adolescence to college [6,61]. Good family functioning can reduce depressive symptoms [62]. Preventive measures like positive communication and promoting healthy habits can improve youth mental health [57]. Positive family cohesion buffers the impact of moral disengagement [63]. Living with family members and having a stable family income are protective factors [9]. Higher quality interactions with schools and mental health services benefit students’ mental health [32]. Physical activity and seeking counseling are protective factors in reducing depressive symptoms and suicide-related ideation [64]. Increased interaction with faculty and staff can prevent adjustment disorders and support social adjustment [61]. Strengthening relationships with counselors can improve the mental health of students studying abroad [65].

College students face academic and employment pressures [3,66], and pandemic-related disruptions affect mental health [67]. Good social support is a protective factor during the pandemic [68]. Social support helps students reduce negative emotions [69] and cope with stressful events [70]. Social support can alleviate depressive symptoms and enhance posttraumatic growth [71].

Therefore, the significance of this research lies in its ability to deepen our understanding of the mental health challenges faced by college students during their transition to adulthood, especially in the context of the COVID-19 pandemic. By exploring how emotional intelligence, loneliness, and social support mediate the relationship between family functioning and mental health, this study provides valuable insights that can inform targeted interventions to enhance student well-being. The findings highlight the importance of fostering supportive environments both within families and educational institutions, ultimately contributing to more effective mental health strategies for young adults navigating this critical developmental stage.

## 2. Objectives and Hypotheses

The following are research hypotheses that are built based on the objectives of the study.

Objective 1: To analyze the difference in depressive symptoms according to demographic factors (gender, academic discipline and child status).

**Ha1.** 
*There is a difference in depressive symptoms according to gender.*


**Ha2.** 
*There is a difference in depressive symptoms according to academic discipline.*


**Ha3.** 
*There is a difference in depressive symptoms based on child status (only children vs. not only children).*


Objective 2: To examine the relationship between family functioning, emotional intelligence, loneliness, and social support on depressive symptoms.

**Ha4.** 
*There is a relationship between family functioning and depressive symptoms.*


**Ha5.** 
*There is a relationship between emotional intelligence and depressive symptoms.*


**Ha6.** 
*There is a relationship between loneliness and depressive symptoms among.*


**Ha7.** 
*There is a relationship between social support and depressive symptoms.*


Objective 3: To analyze whether social support will act as a moderating factor in the relationship between family functioning and depressive symptoms.

**Ha8.** 
*The association between family functioning and depressive symptoms is moderated by social support.*


Objective 4: To analyze whether loneliness and emotional intelligence will act as mediating factors in the relationship between family functioning and depressive symptoms.

**Ha9.** 
*The association between depressive symptoms and family functioning is mediated by emotional intelligence and loneliness.*


## 3. Methods

This study employed a cross-sectional research design to examine the correlation between variables such as family functioning, emotional intelligence, loneliness, social support, and depressive symptoms among college students in China. The data were collected through questionnaires at a single point in time, allowing for a quick and effective method to identify relationships between these variables [72]. The cross-sectional approach is advantageous for collecting data from a large sample and comparing differences between groups, such as males and females.

The research was conducted at Xi’an Jiaotong University (XJTU), one of China’s oldest and most prestigious universities. Established in 1896, XJTU has evolved into a comprehensive research university with nine major disciplines, namely, Science, Engineering, Medicine, Economics, Management, Humanities, Law, Philosophy, and Art. The university has 27 schools, 9 undergraduate colleges, and 3 affiliated hospitals, making it an ideal location for this study.

The target population for this study included undergraduate students at Xi’an Jiaotong University. As of 2023, XJTU has 22,240 undergraduate students, 21,725 master’s students, and 8403 doctoral students. Specifically, the study focused on the 6050 first-year freshmen and the total undergraduate population of 24,051 students (http://www.xjtu.edu.cn/jdgk/tjsj.htm (accessed on 5 January 2024)).

This study utilized a cluster sampling technique, which involves dividing the population into clusters based on shared traits and randomly selecting clusters for inclusion in the sample. XJTU has 13 schools/centers offering undergraduate studies in sciences and 16 in social sciences, as shown in Table 1. The sample was selected using a two-stage cluster sampling technique, with the sample size calculated using the Bukhari sample size calculator. To account for incomplete data, the sample size was increased by 15 percent, resulting in a total of 415 undergraduate students.

Participants were selected based on specific inclusion and exclusion criteria. The inclusion criteria were: (1) active undergraduate students, (2) citizens of China, and (3) proficiency in reading, writing, and typing in Mandarin. The exclusion criteria were: (1) currently receiving medical treatment for a chronic disease, (2) history of mental health disorders, (3) severe cognitive impairment or psychiatric disorders, and (4) unwillingness to provide informed consent. These criteria were assessed through direct questions to ensure appropriate participant selection.
$$Sample\;size=\frac{\frac{z^{2}\;\times \;p(1-p)}{e^{2}}}{1+(\frac{z^{2}\;\times \;p(1-p)}{e^{2}N})}$$

As shown in Figure 1, data were collected through a structured questionnaire that assesses family functioning, emotional intelligence, loneliness, social support, and depressive symptoms. This method ensures that data are gathered consistently and can be analyzed effectively to identify correlations between the variables. The questionnaire was distributed to the selected sample at a single point in time, reflecting the participants’ current states.

The collected data were analyzed using statistical methods to determine the relationships between the variables. Correlation analysis and regression analysis were employed to identify significant correlations. The analysis also explored differences between male and female students to provide a comprehensive understanding of the factors affecting student mental health.

This cross-sectional study at Xi’an Jiaotong University aimed to investigate the relationships between various psychological and social factors and depressive symptoms among college students. Using robust sampling techniques and clear inclusion and exclusion criteria, this research sought to offer valuable insights into the mental health challenges faced by students and inform potential interventions to support their well-being.

## 4. Results

Further analysis was carried out to determine gender differences in depressive symptoms. The null hypothesis (Ho1) is that there is no significant difference in the likelihood of experiencing depression between female and male undergraduate students, while the alternative hypothesis (Ha1) is that there is a significant difference in the likelihood of experiencing depression between female and male undergraduate students. To investigate gender disparities in depressive symptoms, an independent samples *t*-test was performed to compare the depressive symptom scores of male and female participants (see Table 2). The *t*-test revealed a statistically significant difference in depressive symptom scores between males and females (t(df) = −3.691, *p* < 0.01). Specifically, males exhibited a lower mean depressive symptom score (M = 0.498, SD = 0.348) compared to females (M = 0.639, SD = 0.429). The hypothesis that there are differences in depressive symptoms based on gender is supported by the evidence.

The second null hypothesis (Ho2) is that there is no significant difference in depressive symptoms across academic disciplines among undergraduate students, while the alternative hypothesis (Ha2) is that there is a significant difference in depressive symptoms across academic disciplines among undergraduate students. To investigate potential differences in depressive symptoms across academic disciplines, an independent samples *t*-test was conducted. This analysis compared the depressive symptom scores of participants from science-oriented subjects to those studying humanities-based fields. As shown in Table 2, the *t*-test results did not reveal any statistically significant differences in depressive symptom scores between students from these two broad subject areas (t(df) = −0.459, *p* > 0.05). The hypothesis that there are differences in depressive symptoms based on academic disciplines is not supported.

The third null hypothesis (Ho3) is that there is no significant difference in depressive symptoms between only children and non-only children among undergraduate students, while the alternative hypothesis (Ha3) is that there is a significant difference in depressive symptoms between only children and non-only children among undergraduate students. To analyze differences in depressive symptoms between only children and non-only children, an independent samples *t*-test was conducted on their depressive symptom scores (see Table 2). The results of the *t*-test revealed no statistically significant difference in depressive symptom scores between only children and those with siblings, t(df) = −0.36, *p* = 0.717, 95% CI [−0.11, 0.07]. This suggests that being an only child is not associated with higher levels of depressive symptoms compared to having siblings in this sample. The hypothesis that there is no significant difference in depressive symptoms between only children and non-only children among undergraduate students is supported.

No outliers were identified using the outlier labeling rule (Hoaglin & Iglewicz 1987). Bivariate Pearson’s correlations were calculated to explore the relationships between family functioning, emotional intelligence, loneliness, social support, and depressive symptoms (Table 3).

For the relationship between family functioning and depressive symptoms, the null hypothesis (Ho4) states that there is no significant relationship, while the alternative hypothesis (Ha4) suggests there is a significant relationship. The results indicated a significant relationship between family functioning and depressive symptoms among undergraduate students (Table 3).

For the relationship between emotional intelligence and depressive symptoms, the null hypothesis (Ho5) states that there is no significant relationship, while the alternative hypothesis (Ha5) suggests there is a significant relationship. The results showed a significant relationship between emotional intelligence and depressive symptoms among undergraduate students (Table 3).

For the relationship between loneliness and depressive symptoms, the null hypothesis (Ho6) states that there is no significant relationship, while the alternative hypothesis (Ha6) suggests there is a significant relationship. The results demonstrated a significant relationship between loneliness and depressive symptoms among undergraduate students (Table 3).

For the relationship between social support and depressive symptoms, the null hypothesis (Ho7) states that there is no significant relationship, while the alternative hypothesis (Ha7) suggests there is a significant relationship. The results showed a significant relationship between social support and depressive symptoms among undergraduate students (Table 3).

### Path Analysis

Following the correlation analysis, a hierarchical regression analysis (stepwise) was performed to determine the importance of family functioning and social support for depressive symptoms (Table 4). Gender (male, female) was treated as a covariate, family functioning as an independent variable, depressive symptoms as a dependent variable, and social support as a moderating variable. Due to the exploratory nature of this research question, variables with non-significant effects were removed stepwise. The hierarchical regression analysis was conducted in three steps to determine the predictive power of these variables on depressive symptoms, as measured by the depressive symptoms scale. Initially, gender was incorporated as a covariate in Step 1 to control its influence, as preliminary analysis suggested potential gender differences in depressive symptoms (Model 1). In Step 2, key psychosocial variables such as family functioning were added to investigate their additional explanatory power (Model 2). Finally, in Step 3, social support and the interaction between family functioning and social support (FF × SS) were added to examine their further explanatory power (Model 3).

The hierarchical regression produced three models. Model 1 introduced gender as a predictor, explaining a modest 3 percent of the variance in depressive symptoms (Adjusted R^2^ = 0.03), which was statistically significant, indicating an initial valid model setup. When Model 2 incorporated family functioning and social support, the explained variance increased to 11.2 percent (Adjusted R^2^ = 0.105), significantly enhancing the model’s explanatory power, as evidenced by the F change and its significance. Model 3 added the interaction term FF × SS, further raising the explained variance to 13.2 percent (Adjusted R^2^ = 0.123). The significant F change highlighted the importance of the interaction term in the regression equation, underscoring the moderating effect of social support.

The F-statistics provided in the ANOVA table validated the model improvements at each step, with all models achieving statistical significance (*p* < 0.001), reflecting robust model fits and justifying the inclusion of additional predictors and the interaction term in successive models.

The coefficients table from Model 3 offers a detailed view of the impact of each predictor. Family functioning showed a significant negative coefficient, indicating that better family functioning is associated with reduced depressive symptoms. Social support also presented a significant negative effect, suggesting that higher levels of social support are associated with fewer depressive symptoms. Crucially, the interaction term FF × SS was positively significant (B = 0.092, *p* = 0.002), illustrating that the effect of family functioning on depressive symptoms is moderated by social support, enhancing its protective effect against depressive symptoms.

The diagnostics for residuals, including the histogram and the normal P-P plot, confirmed the assumption of normality in the distribution of residuals. The histogram exhibited a fairly normal distribution around the mean, which is close to zero, and the P-P plot closely aligned with the diagonal line, indicating a good fit between the observed values and those predicted by the model.

These results collectively confirm that the moderating role of social support significantly impacts the relationship between family functioning and depressive symptoms. The statistical significance of the interaction term substantiates the hypothesis that social support does not merely additively reduce depressive symptoms but also enhances the positive effects of functional family dynamics.

The hierarchical regression analysis conclusively demonstrates that social support moderates the relationship between family functioning and depressive symptoms among the studied population. The significant interaction term highlights that the effectiveness of family functioning in mitigating depressive symptoms is contingent upon the level of social support, supporting the alternative hypothesis (Ha8) over the null hypothesis (Ho8).

As shown in Figure 2, further analysis was conducted to determine if emotional intelligence and loneliness act as mediators in the relationship between family functioning and depressive symptoms. As shown in Figure 3, the blue dots represent the observed cumulative distribution of the residuals, while the diagonal line represents the expected values under a normal distribution. The blue dots closely follow the diagonal line, indicating that the residuals are approximately normally distributed, supporting the model’s assumption of normality.

The null hypothesis (Ho9) states that the association between depressive symptoms and family functioning is not mediated by emotional intelligence and loneliness. The alternative hypothesis (Ha9) suggests that this association is indeed mediated by these factors.

To explore this complex relationship, a hierarchical regression analysis using the PROCESS Macro by Andrew Hayes was performed. This analysis examined family functioning as an independent variable and depressive symptoms as the dependent variable, with emotional intelligence and loneliness as mediators, and gender as a covariate. The results were presented across multiple models, reflecting both direct and indirect effects within the proposed mediation framework.

Model 1 investigated the relationship between family functioning and emotional intelligence, controlling for gender. The model was significant (R^2^ = 0.0280, F(2, 412) = 5.9427, *p* = 0.0029), with family functioning explaining 2.8% of the variance in emotional intelligence (B = 0.2582, SE = 0.0779, *t* = 3.3162, *p* = 0.0010). Gender was not a significant predictor (B = −0.0763, *p* = 0.3886).

Model 2 explored the effects of family functioning and emotional intelligence on loneliness, again controlling for gender. This model was also significant (R^2^ = 0.0556, F(3, 411) = 8.0592, *p* < 0.0001), explaining 5.56% of the variance in loneliness. Emotional intelligence significantly predicted lower loneliness (B = −0.0900, SE = 0.0318, *t* = −2.8276, *p* = 0.0049). Family functioning had a marginally non-significant effect on loneliness (B = −0.0865, SE = 0.0510, *t* = −1.6966, *p* = 0.0905), while gender was a significant predictor (B = 0.1832, *p* = 0.0015).

Model 3 assessed the direct and indirect effects of family functioning on depressive symptoms, with emotional intelligence and loneliness as mediators. The model was significant (R^2^ = 0.3057, F(4, 410) = 45.5182, *p* < 0.0001), explaining 30.57% of the variance in depressive symptoms. Family functioning was a significant negative predictor of depressive symptoms (B = −0.1299, SE = 0.0290, *t* = −4.4807, *p* < 0.0001), while loneliness was a strong positive predictor (B = 0.3023, SE = 0.0280, *t* = 10.8120, *p* < 0.0001). Emotional intelligence approached significance (B = −0.0307, SE = 0.0182, *t* = −1.6858, *p* = 0.0926), and gender was also significant (B = 0.0763, *p* = 0.0205). The mediation effect model is shown in Figure 4.

As shown in Figure 4, ss for direct and indirect effects of family functioning on depressive symptoms, the direct effect of family functioning on depressive symptoms was significant (B = −0.1299, SE = 0.0290, t = −4.4807, *p* < 0.0001), suggesting that family functioning has a direct, negative effect on depressive symptoms, independent of the mediators. The total indirect effect of family functioning on depressive symptoms through the mediators was significant (Indirect Effect = −0.0411, BootSE = 0.0176, 95% CI [−0.0767, −0.0066]), indicating that the combined mediation pathways account for a significant portion of the effect of family functioning on depressive symptoms. Specific indirect effects are as follows: The first pathway (family functioning → emotional intelligence → depressive symptoms) suggests that family functioning impacts depressive symptoms through emotional intelligence (Indirect Effect = −0.0079, BootSE = 0.0059, 95% CI [−0.0213, 0.0015]). The second pathway (family functioning → loneliness → depressive symptoms) indicates the mediation role of loneliness in the relationship between family functioning and depressive symptoms (Indirect Effect = −0.0261, BootSE = 0.0150, 95% CI [−0.0560, 0.0031]). The third pathway (family functioning → emotional intelligence → loneliness → depressive symptoms) demonstrates a sequential mediation where emotional intelligence mediates between family functioning and loneliness, which in turn affects depressive symptoms (Indirect Effect = −0.0070, BootSE = 0.0038, 95% CI [−0.0158, −0.0011]). The analysis of the chain mediation effect is shown in Table 5.

To sum up, the findings indicate that while the overall indirect effect of family functioning on depressive symptoms is significant, the specific mediation effects through emotional intelligence and loneliness individually are not. However, the significant sequential mediation effect underscores the complexity of these relationships, highlighting the importance of considering sequential mediation in psychological research.

## 5. Discussion

The gender differences observed in this study highlight a significant disparity in depressive symptoms between female and male undergraduate students. This result aligns with prior research indicating that female students are more likely to experience depression than their male counterparts [2,73,74,75]. The reasons behind this gender difference may be complex and include biological bases, psychological characteristics, and socio-environmental disturbances [76]. For example, it has been shown that women tend to have higher caregiving responsibilities, which may explain the higher risk for anxiety and depression among women [77,78,79]. Additionally, women tend to suffer more from internalizing disorders like depression due to factors like brain structure, genetic influences, and hormonal fluctuations, while men are more prone to externalizing disorders like substance abuse [80]. Gender discrimination and motherhood discrimination also contribute to the higher incidence of mental disorders in women [81,82,83,84,85].

Interestingly, this study found no significant differences in depressive symptoms across subjects. Recent studies have supported this perspective, highlighting those common stressors faced by all students, such as academic workload, financial challenges, social isolation, and the overall adjustment to university life, may play a more substantial role in the development of depressive symptoms than previously recognized. For example, a study by Jumani emphasized that health sciences students are particularly vulnerable due to the general academic pressures rather than discipline-specific factors [86]. Similarly, network analysis by Wang suggested that while certain disciplines may have unique stressors, these do not necessarily lead to variations in depressive symptoms across different academic fields [87]. However, the uniformity in depressive symptoms across disciplines does not negate the possibility that the nature and sources of stress might differ between fields. For example, recent research indicates that students in high-stakes disciplines, such as medicine and engineering, continue to experience unique stressors that could potentially lead to burnout or anxiety, though these do not necessarily manifest as depressive symptoms. A study explored the relationship between negative life events and depressive symptoms, suggesting that while these students are at risk for various mental health challenges, they might experience these as burnout or anxiety rather than depression [88]. Additionally, the implementation of broad mental health support systems by universities, which are accessible to all students, might also contribute to the uniformity in depressive symptoms observed. These resources, such as counseling services, mental health workshops, and peer support networks, provide a buffer against the development of depressive symptoms that could be attributed to academic pressures specific to certain fields [89].

Additionally, this study found no significant differences in depressive symptoms between only children and non-only children in undergraduate programs. This finding contrasts with previous research suggesting that only children may be less susceptible to depression. For instance, Tong argue that families with only children often provide more concentrated parental attention and resources, which fosters emotional stability and resilience in these children, potentially enhancing their ability to cope with stress effectively [90]. Conversely, other studies have suggested that non-only children might be less susceptible to depression due to the support provided by siblings.R researchers found that sibling relationships can buffer the negative effects of family conflict by offering emotional support, which helps mitigate depressive symptoms during times of stress [91]. The inconsistencies in these findings may be attributed to sample limitations and cultural differences. For example, some studies highlighted that social norms surrounding family size can significantly influence mental health outcomes, with varying cultural expectations potentially affecting the development of depressive symptoms in children [92]. Therefore, further comprehensive and cross-cultural comparative studies are necessary to better understand this complex phenomenon.

Overall, depression among college students requires attention and intervention. The lack of significant differences between only and non-only students, across disciplines, indicates that depressive symptoms are evenly distributed across the undergraduate population. This could mean that the university environment has a homogenizing effect on students’ mental health, masking other demographic variables. This consistency suggests that interventions and support services should be equally accessible and replicable across all gender groups and academic levels, regardless of whether students are only children.

The study showed that family functioning, emotional intelligence, and social support were significantly and negatively related to depressive symptoms. In contrast, loneliness was positively associated with depressive symptoms.

Family functioning was significantly negatively associated with depressive symptoms. The quality of family interactions plays a crucial role in mental health outcomes [93,94,95,96]. Effective communication, emotional support, and cohesive family relationships are crucial in reducing depressive symptoms. Negative family dynamics can exacerbate depression, suggesting that family interactions can mitigate or amplify the effects of external stressors on mental health [97]. Additionally, good family functioning may reduce the likelihood of adverse childhood events and financial difficulties, lowering the probability of depressive symptoms among college students [98]. These findings have important implications for targeted mental health interventions. Universities and mental health professionals should focus on enhancing family functioning through family-based interventions and counseling services.

The strong negative correlation between emotional intelligence and depressive symptoms emphasizes the protective role of emotional intelligence in mental health. Students with higher emotional intelligence are better able to manage their emotions and cope with stress, reducing their risk of depression [99,100]. Individuals with high emotional intelligence demonstrate superior emotion regulation, handle stress and negative emotions more effectively, and are less likely to develop depressive symptoms [101]. People with high emotional intelligence are also better at recognizing and responding to emotional cues, contributing to better interpersonal relationships and social support networks, which are crucial for reducing feelings of isolation and loneliness [102]. Emotional intelligence training significantly reduces depression severity [103,104,105]. These findings advocate for the integration of emotional intelligence training in mental health interventions.

The positive correlation between loneliness and depressive symptoms aligns with research suggesting that social isolation and loneliness are significant risk factors for depression [106,107]. Loneliness exacerbates depressive symptoms by increasing negative cognitive biases and reducing opportunities for positive social interactions [108,109]. Loneliness is associated with higher levels of stress and negative emotions, which can directly contribute to depression [110]. The bidirectional relationship between loneliness and depression suggests a cycle where each condition worsens the other [111]. These findings highlight the importance of reducing loneliness through social integration programs and peer support networks.

The negative correlation between social support and depressive symptoms is well supported by existing literature [103,112,113]. Social support acts as a protective factor that mitigates the negative effects of stress on mental health. High social support buffers the effects of stress on depression, while low social support exacerbates this relationship. Emotional support promotes psychological well-being, and social companionship can help reduce depression [114,115]. Lack of social support can increase feelings of hopelessness and isolation, potentially leading to suicide risk [116]. Longitudinal studies provide further insight into this dynamic relationship, showing that initial social support predicts fewer depressive symptoms later on [117]. This bidirectional relationship suggests a cycle where enhanced social support leads to better mental health, which in turn helps maintain and strengthen social networks. The strong relationship between social support and depressive symptoms underscores the importance of fostering strong social networks as a strategy for preventing and treating depression.

The results of the study suggest that the level of social support does influence the strength and direction of the relationship between family functioning and depressive symptoms. These results are consistent with the existing literature, which suggests that the protective effects of family functioning on depressive symptoms are stronger for students who perceive high levels of social support. Conversely, for students with low levels of social support, the beneficial impact of family functioning on reducing depressive symptoms is weakened [115,118,119]. This interaction highlights the importance of considering both family dynamics and external social support when addressing mental health issues. Several recent studies further elucidate these mechanisms. For instance, a study found that social support significantly moderated the relationship between family functioning and depressive symptoms, with high social support buffering against the negative effects of poor family functioning [120]. Another study highlighted that perceived social support reduced the impact of stress on depressive symptoms, emphasizing its protective role [121]. Research has shown that social support helps individuals manage stress more effectively, which in turn reduces depressive symptoms [120].

The joint mediating effects of emotional intelligence and loneliness reveal the complex mechanisms behind mental health, particularly in students’ social and emotional lives. The analysis shows that students from supportive family environments develop higher emotional intelligence, which helps them manage stress and regulate emotions, thus reducing depression risk [122,123,124]. However, emotional intelligence alone did not significantly mediate depressive symptoms, suggesting that its impact may depend on other factors like loneliness. Further analysis indicated that positive family interactions improve interpersonal relationships and reduce loneliness, leading to fewer depressive symptoms [122]. Loneliness is well known to exacerbate negative cognitive biases and stress responses, contributing to depression. Thus, while emotional intelligence and loneliness may not independently mediate depressive symptoms, their combined role in the sequential mediation pathway “Family Functioning → Emotional Intelligence → Loneliness → Depressive Symptoms” is significant. This suggests that improving family dynamics can enhance emotional intelligence, reduce loneliness, and more effectively alleviate depressive symptoms [103,113,114]. 

## 6. Conclusions and Limitations

### 6.1. Conclusions

This research investigated the associations between family functioning, emotional intelligence, loneliness, social support, and depressive symptoms among college students in China. The study utilized a cross-sectional design, employing a variety of validated instruments to measure the variables of interest. The main findings underscored the significant roles of family functioning, emotional intelligence, and social support in mitigating depressive symptoms, while loneliness emerged as a critical risk factor for depression.

The participants were drawn from a diverse population of undergraduate students, ensuring a broad representation of different demographic factors. The data collection process was meticulously designed to capture the multifaceted nature of mental health among students, focusing on both individual and environmental factors that contribute to depressive symptoms. This comprehensive approach allowed for a detailed analysis of the complex interactions between the studied variables.

The findings revealed that family functioning plays a crucial role in shaping students’ emotional intelligence and social support networks, which in turn influence their levels of loneliness and depressive symptoms. High levels of family functioning were associated with better emotional intelligence and stronger social support, which acted as protective factors against depression. Conversely, students who reported lower levels of family functioning were more likely to experience higher levels of loneliness and depressive symptoms.

### 6.2. Limitations

This subsection discusses the limitations of this study in detail. We honestly explore the issues and limitations that were encountered during the research process and describe the impact that these factors may have on the interpretation and generalization of the research results.

Firstly, the cross-sectional design limits the ability to establish causal relationships between the variables. While the study provides valuable insights into the associations between family functioning, emotional intelligence, loneliness, social support, and depressive symptoms, it cannot definitively determine the directionality of these relationships. Longitudinal studies are needed to explore how these relationships evolve over time and to establish causality.

Additionally, the sample was drawn from a specific population of college students in China, which may limit the generalizability of the findings to other populations or cultural contexts. The unique cultural, social, and educational environment in China may influence the relationships between the studied variables differently than in other contexts. Future research should include diverse samples from different cultural and educational settings to provide a broader understanding of these relationships.

## Figures and Tables

**Figure 1 behavsci-14-00819-f001:**
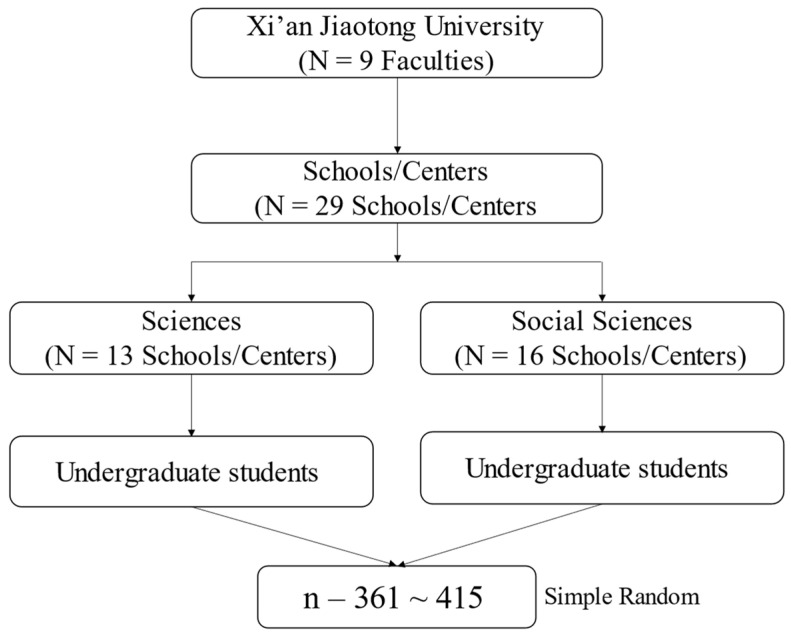
Cluster sampling, two stages.

**Figure 2 behavsci-14-00819-f002:**
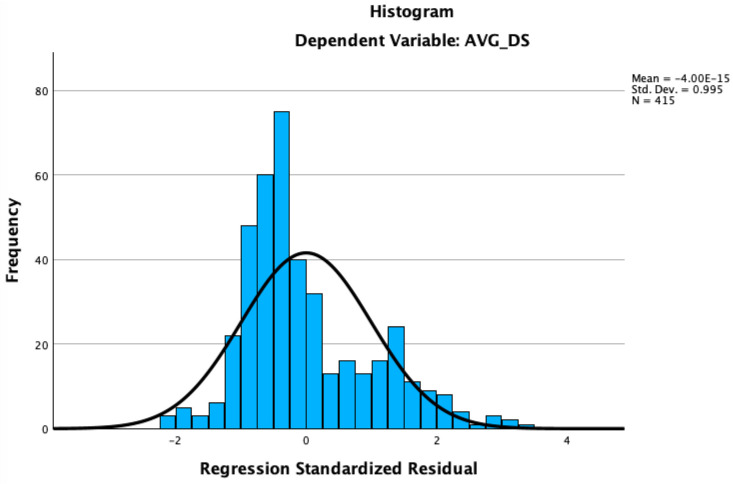
Histogram of Regression Standardized Residual.

**Figure 3 behavsci-14-00819-f003:**
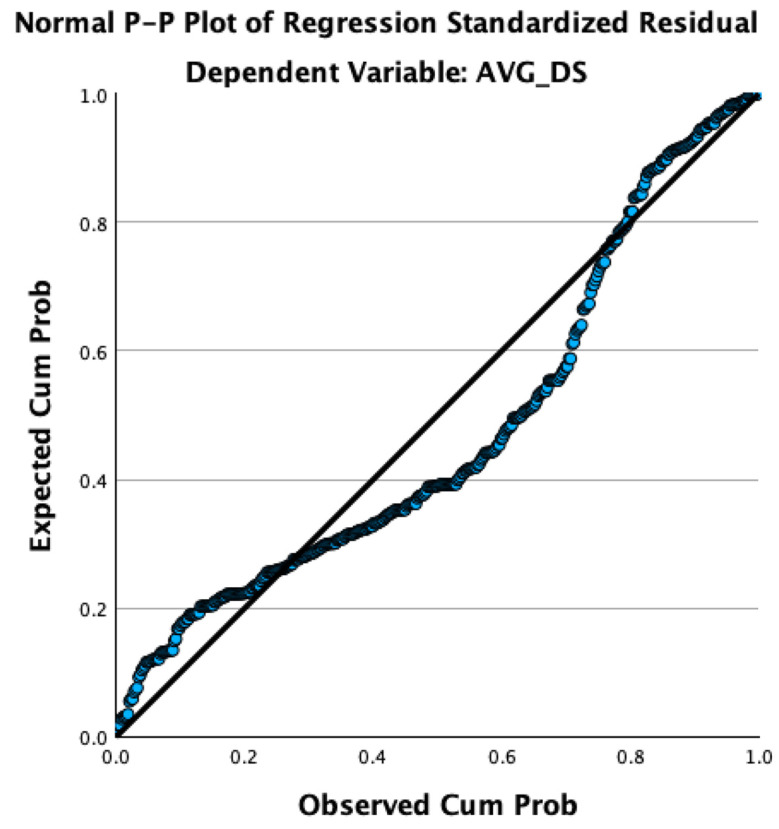
Normal P-P plot of regression standardized residual.

**Figure 4 behavsci-14-00819-f004:**
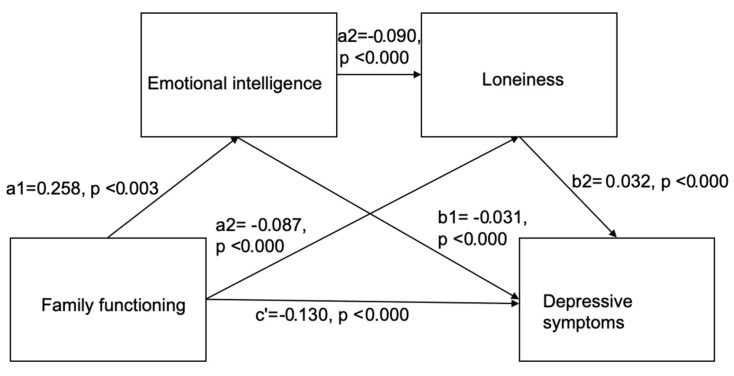
Mediation model.

**Table 1 behavsci-14-00819-t001:** Schools and centers of Xi’an Jiaotong University.

**Sciences**
1. School of Mathematics and Statistics
2. School of Physics
3. School of Chemistry
4. Frontier Institute of Science and Technology
5. School of Mechanical Engineering
6. School of Electrical Engineering
7. School of Energy and Power Engineering
8. Faculty of Electronic and Information Engineering
9. School of Materials Science and Engineering
10. School of Human Settlements and Civil Engineering
11. The School of Life Science and Technology
12. School of Aerospace Engineering
13. School of Chemical Engineering and Technology
**Social Sciences**
1. School of Economics and Finance
2. Jinhe Center For Economic Research
3. School of Management
4. School of Public Policy and Administration
5. School of Humanities and Social Science
6. School of Journalism and New Media
7. School of Marxism Studies
8. School of Law
9. The School of Foreign Studies
10. Center of Physical Education
11. School of Continuing Education
12. Distant Education School
13. School of International Education
14. Innovation and Entrepreneurship School
15. XJTU-POLIMI Joint School of Design and Innovation
16. School of Future Technology

**Table 2 behavsci-14-00819-t002:** Between-group gender, subject, and only child differences in depressive symptoms among undergraduates.

Characteristics		Number of Participants (n)	Mean ± Standard Deviation	*t*
Gender	Male	233	0.498 ± 0.348	−3.691 **
	Female	182	0.639 ± 0.429	
Subject	Science	207	0.551 ± 0.369	−0.459
	Humanities	208	0.568 ± 0.413	
Only Child	Yes	325	0.556 ± 0.396	−0.363
	No	90	0.573 ± 0.375	

Note: “**” indicates *p* < 0.01.

**Table 3 behavsci-14-00819-t003:** The relationship between studied variables and depressive symptoms.

Variables	Depressive Symptoms r
FF	−0.25 **
EI	−0.18 **
SEA	−0.13 **
OEA	−0.18 **
ROE	−0.17 **
UOE	−0.17 **
Loneliness	0.50 **
Social support	−0.18 **
Family support	−0.16 **
Friends’ support	−0.18 **
Others support	−0.17 **

Note: “**” indicates *p* < 0.01.

**Table 4 behavsci-14-00819-t004:** Hierarchical regression analysis (stepwise): the relationship between family functioning, social support, and depressive symptoms (N = 415).

Factors		Unstandardized Coefficients		Standardized Coefficients Beta	*t*	Sig.	Collinearity Statistics	
		B	Std. Error				Tolerance	VIF (Variance Inflation Factor)
Model 1	(Constant)	0.36	0.06		6.15	<0.001		
	gender	0.14	0.04	0.18	3.70	<0.001	1.00	1.00
ΔR^2^ (Change in R-Squared): 0.032, F (1, 413) = 13.620, *p* < 0.001								
Model 2	(Constant)	0.95	0.11		8.45	<0.001		
	gender	0.13	0.04	0.17	3.56	<0.001		
	family functioning	−0.16	0.03	−0.22	−4.76	<0.001	0.991	1.009
	social support	−0.06	0.02	−0.14	−2.96	0.003	0.991	1.009
ΔR^2^ (Change in R-Squared): 0.112 − 0.032 = 0.080, F (3, 411) = 17.208, *p* < 0.001								
Model 3	(Constant)	1.64	0.25		6.53	<0.001		
	gender	0.13	0.04	0.16	3.48	<0.001		
	family functioning	−0.42	0.092	−0.61	−4.55	<0.001	0.92	1.09
	social support	−0.30	0.08	−0.69	−3.72	<0.001	0.92	1.09
	FF × SS	0.09	0.03	0.73	3.07	0.002	0.99	1.02
ΔR^2^ (Change in R-Squared): 0.132 − 0.112 = 0.020, F (4, 410) = 15.521, *p* < 0.001								

Note: FF × SS (Interaction between family functioning and social support).

**Table 5 behavsci-14-00819-t005:** An analysis of the chain mediating effect of emotional intelligence and loneliness.

Indirect Paths	Indirect Effect	BootSE	95% CI
Family functioning → emotional intelligence → depressive symptoms	−0.0079	0.0059	[−0.0213, 0.0015]
Family functioning→ loneliness → depressive symptoms	−0.0261	0.0150	[−0.0565, 0.0031]
Family functioning → emotional intelligence → loneliness → depressive symptoms	−0.0070	0.0038	[−0.0158, −0.0011]
Total	−0.0411	0.0176	[−0.0767, −0.0066]

## Data Availability

No new data were created or analyzed in this study. Data sharing is not applicable to this article.

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
