# Peer review of "The Relationship between Family Functioning, Emotional Intelligence, Loneliness, Social Support, and Depressive Symptoms among Undergraduate Students"

_behavsci, 2024, doi:10.3390/bs14090819_

Round 1

Reviewer 1 Report

Comments and Suggestions for Authors

This research investigates the association between family functioning, emotional intelligence, loneliness, social support, and depressive symptoms in Chinese college students. The study used A cross-sectional design and a sample from Xi'an Jiaotong University. The study found that family functioning, emotional intelligence, and social support were negatively correlated with depressive symptoms. Also, loneliness was positively correlated with depressive symptoms. Moreover, social support significantly moderated the relationship between family functioning and depressive symptoms. Finally, emotional intelligence and loneliness mediated the relationship.

  1. The topic has to be edited. It states, "The relationship of Family Functioning, Emotional Intelligence, Loneliness and Social Support on Depressive Symptoms Among Undergraduate Students ". It should rather be 'The relationship between Family Functioning, Emotional Intelligence, Loneliness and Social Support on Depressive Symptoms Among Undergraduate Students'. Alternatively, it could be stated as 'The effect of Family Functioning, Emotional Intelligence, Loneliness and Social Support on Depressive Symptoms Among Undergraduate Students.
  2. The researchers should have spelled out the purpose of the study.
  3. The paper contained no research questions or hypotheses.
  4. An important ingredient missing in this paper is that the researchers did not inform readers why this research is significant. The authors have to make a case for this research.
  5. The authors did not mention the weaknesses of the study.
  6. The discussion section on the mediation effect of loneliness and emotional intelligence should be edited because it contains several repetitions of ideas.
Comments on the Quality of English Language

Some editing is needed in this manuscript.

Author Response

1.The topic has to be edited. It states, "The relationship of Family Functioning, Emotional Intelligence, Loneliness and Social Support on Depressive Symptoms Among Undergraduate Students ". It should rather be 'The relationship between Family Functioning, Emotional Intelligence, Loneliness and Social Support on Depressive Symptoms Among Undergraduate Students'. Alternatively, it could be stated as 'The effect of Family Functioning, Emotional Intelligence, Loneliness and Social Support on Depressive Symptoms Among Undergraduate Students.

Response 1:

Thank you for pointing this out. I agree on this comments. The revised contents can be found in line 2-4.

2.The researchers should have spelled out the purpose of the study.

Response 2: 
Thank you for pointing this out. I agree on this comments. The revised contents can be found in line 157-196.

3.The paper contained no research questions or hypotheses.
Response 3: 
Thank you for pointing this out. I agree on this comments. The revised contents can be found in line 157-196.

4.An important ingredient missing in this paper is that the researchers did not inform readers why this research is significant. The authors have to make a case for this research.
Response 4:

Thank you for pointing this out. I agree on this comments. The revised contents can be found in line 147-156.

5.The authors did not mention the weaknesses of the study.
Response 5:

Thank you for pointing this out. I agree on this comments. The revised contents can be found in line 626-643.

6.The discussion section on the mediation effect of loneliness and emotional intelligence should be edited because it contains several repetitions of ideas.
Response 6:

Thank you for pointing this out. I agree on this comments. The revised contents can be found in line 387-443.

Reviewer 2 Report

Comments and Suggestions for Authors

dear authors congratulation to your research article

The relationship of Family Functioning, Emotional Intelligence, Loneliness and Social support on Depressive Symptoms Among Undergraduate Students 

Content:

This study describes the relationships between family functioning, emotional intelligence, loneliness, social support, and depressive symptoms in Chinese college students

1. Well - structured paper 

2. Authors understand this topic. 

3. excellent ideas line  390 - 392 

women tend to suffer more from internalizing disorders like depression due to factors like brain structure, genetic influences, and hormonal fluctuations, while men are more prone to externalizing disorders like substance abuse ...

4. The authors have a clear structure and ideas of the article, including a logical progression

5. The authors explain the problems very concretely 

tables - discussion -conclusion  - no problem 

it is great idea of the authors: :  emotional Intelligence, loneliness  and how is important:  social support on depressive symptoms 

recommandation: 

 1. Background  /  1. Introducing 

2, in the text is only 3 times mentioned "anxiety 66, 75, 389) 

it is necessary to explain more about the "concept of anxiety"

Pavlikova, M., & Tavilla, I. (2023). Repetition as a Path to Authentic Existence in Kierkegaard’s Work. Journal of Education Culture and Society14(2), 105-115. https://doi.org/10.15503/jecs2023.2.105.115 

3 the problem of covid / anxiety / 

 the idea of covid in connection with Anxiety  needs more explanation 

Percent match: 18% : 

18 percents of citations is ok. 

Conclusion: 

I recommend this article to publish in this journal. 

Author Response

1.Background  /  1. Introducing 
Response 1:

Thank you for pointing this out. I agree on this comment. The revised contents can be found in line 34.

2, in the text is only 3 times mentioned "anxiety 66, 75, 389) 
it is necessary to explain more about the "concept of anxiety"
Pavlikova, M., & Tavilla, I. (2023). Repetition as a Path to Authentic Existence in Kierkegaard’s Work. Journal of Education Culture and Society, 14(2), 105-115. https://doi.org/10.15503/jecs2023.2.105.115 
Response 2:

Thank you for pointing this out. I agree on this comment. The revised contents can be found in line 824.

3 the problem of covid / anxiety / 
 the idea of covid in connection with Anxiety  needs more explanation 
Response 3:

Thank you for pointing this out. I agree on this comment. The revised contents can be found in line 913.
